# Population Response of Rhizosphere Microbiota of Garden Pea Genotypes to Inoculation with Arbuscular Mycorrhizal Fungi

**DOI:** 10.3390/ijms24021119

**Published:** 2023-01-06

**Authors:** Slavka Kalapchieva, Ivanka Tringovska, Radka Bozhinova, Valentin Kosev, Tsveta Hristeva

**Affiliations:** 1Maritsa Vegetable Crops Research Institute, Agricultural Academy, 4003 Plovdiv, Bulgaria; 2Tobacco and Tobacco Products Institute, Agricultural Academy, 4108 Plovdiv, Bulgaria; 3Institute of Forage Crops, Agricultural Academy, 5800 Pleven, Bulgaria

**Keywords:** *Pisum sativum* L., arbuscular mycorrhizal fungi, trophic groups of soil microorganisms, microbial communities, rhizosphere

## Abstract

This study of a legume’s rhizosphere in tripartite symbiosis focused on the relationships between the symbionts and less on the overall rhizosphere microbiome. We used an experimental model with different garden pea genotypes inoculated with AM fungi (*Rhizophagus irregularis* and with a mix of AM species) to study their influence on the population levels of main trophic groups of soil microorganisms as well as their structure and functional relationships in the rhizosphere microbial community. The experiments were carried out at two phenological cycles of the plants. Analyzes were performed according to classical methods: microbial population density defined as CUF/g a.d.s. and root colonization rate with AMF (%). We found a proven dominant effect of AMF on the densities of micromycetes and actinomycetes in the direction of reduction, suggesting antagonism, and on ammonifying, phosphate-solubilizing and free-living diazotrophic Azotobacter bacteria in the direction of stimulation, an indicator of mutualistic relationships. We determined that the genotype was decisive for the formation of populations of bacteria immobilizing mineral NH_4_^+^-N and bacteria Rhizobium. We reported significant two-way relationships between trophic groups related associated with soil nitrogen and phosphorus ions availability. The preserved proportions between trophic groups in the microbial communities were indicative of structural and functional stability.

## 1. Introduction

The green garden pea (*Pisum sativum* L.) is one of the main protein food crops and its production on a world scale is increasing progressively [1,2]. The importance of the pea crop, being a part of the legume family to the systems of organic and sustainable agriculture [3,4], determines the need for a full inclusion of bioproducts as an alternative to agrochemicals in the technologies for its cultivation. Bioproducts developed based on arbuscular mycorrhizal fungi [5,6,7,8,9] with proven multifaceted benefits in agroecological aspects are rapidly entering agricultural practices [10,11,12,13,14]. Arbuscular mycorrhizal fungi (AMF) are obligate biotrophic soil microorganisms which, in order to complete their life development, form symbiotic relationships with over 90% of surface plants [15,16,17]. This type of symbiosis is thought to be the progenitor (small-genome AM, probably of the order Paraglomerales) of other types of symbiotic relationships with plants [18,19,20]. Peas successfully form symbiotic relationships with two microbial symbionts: AM fungi and the diazotrophic root-nodulating bacteria, Rhizobium [21,22,23]. A significant part of the research on the tripartite symbiosis in legumes focused on the synergistic effects of the two symbionts on the growth, development and nutrition of plants [5,6,7,24,25,26] the processes of nodulation and colonization [8,22,27] and on signaling mechanisms between partners [9]. The establishment of a successful tripartite symbiosis results from bidirectional processes of signaling and exchange of bioactive molecules between plants [28,29,30], microbial partners [31,32,33,34] and a series of events from molecular to the organ level controlled by the so-called “common pathway of symbiosis” [23,35,36,37,38,39].

These communications take place in the rhizospheres of these plants. The rhizosphere is the space in the soil environment directly affected by root secretions. In this part of the soil ecosystem, plant roots, soil and soil biota interact with each other. The rhizosphere is a unique ecological niche inhabited by a microbial population of enormous scale and diversity [40,41,42] where continuous biochemical processes related to the general biochemical cycle of substances take place [43].

Different plants form specific microbial communities [40] which are characterized by heterogeneity and mosaicism on one hand and strictly regulated relationships between the inhabitants and the plants that formed them on the other [44]. These relationships are based on a complex network of plant–microbe–microbe–plant multidirectional biochemical signals coordinated by a process related to the life strategies of microorganisms [45,46]. This process, called “Quorum-Sensing”, reflects the ability of microorganisms to respond to signal molecules that inform them of the current state and any change in their immediate environment caused by biotic and/or abiotic factors. This process was believed to control all types of interactions in the rhizosphere [47,48]. Some authors have compared the rhizosphere area as the stage of a “talking show” with simultaneous discussions and dialogues between multiple participants [49,50].

AM fungi and rhizobia bacteria are important constituents of rhizosphere not only of legumes [51,52]. Moreover, AM fungi form a common subterranean micellar network colonizing various plant taxa, which affects a number of processes in the soil environment and the overall functioning of the ecosystem [27,53]. Their presence directly or indirectly affects the rhizosphere’s microbial communities, their structure, biodiversity, population density and function. On the other hand, the legume–rhizobia symbiosis itself is distinguished by strict specificity between plants (incl. different cultivars) and bacterial species and strains. Plant root secretions, nutrient flow and soil composition are decisive in the formation of specific microbial communities by maintaining a state of homeostasis and optimal functioning [40,49]. How the introduction of AMF inoculants foreign to the already distinct autochthonous microbiota affects the functional relationships between the remaining microbial inhabitants and their abundance is a question not yet fully clarified and/or with conflicting answers [27,48,50,54,55].

Here, we used an experimental model with different garden pea genotypes inoculated with AM fungi (*Rhizophagus irregularis* and with a mix of AM species) to study their influence on the population levels of main trophic groups of soil microorganisms as well as their structure and functional relationships in the rhizosphere microbial community. We found a proven dominant effect of AMF on the densities of micromycetes and actinomycetes in the direction of reduction, suggesting antagonism, and on ammonifying, phosphate-solubilizing and free-living diazotrophic Azotobacter bacteria in the direction of stimulation, an indicator of mutualistic relationships. We determined that the genotype was decisive for the formation of the populations of bacteria immobilizing mineral NH_4_^+^-N and bacteria Rhizobium. We reported significant two-way dependencies between trophic groups of microorganisms, whose vital activity is related to the transformations of nitrogen and phosphorus compounds in the soil. The preserved proportions between trophic groups in the microbial communities were indicative of structural and functional stability.

## 2. Results

For a more complete understanding of the processes taking place in the rhizosphere system, it is increasingly recommended that research on its microbiota be conducted in the complex of natural coenoses [56,57]. The fact is that, very often, results obtained from experiments in precisely controlled conditions are not comparable to or not confirmed with those obtained under field conditions [4,8,55]. In agreement with this conception, the conditions for the performance of our experiments were kept as close as possible to the conditions and technology of growing garden peas in practice (description in Materials and Methods). To analyze the population’s biodiversity, structural and functional stability of the rhizosphere microbiota after inoculation of AM fungi into the rhizosphere of different garden pea genotypes, we selected a network of indicative ecological-trophic soil microorganisms.

The study focused on changes in the population density of the selected indicator groups of microorganisms under the influence of two factors: inoculation with AM fungi (mono-mycorrhizal (mM) and poly-mycorrhizal (pM) products) and different pea genotypes, grouped as “normal” and “afila” types. The division of pea genotypes into two groups was based on their morphological, physiological, economic and other characteristics adopted in the selection of garden peas. The main difference between the plants is the type of leaf. The “afila” type is a new generation of pea varieties with contrasting architectonics of the leaf apparatus and root system. They are distinguished by higher resistance to lodging and weaker photosynthetic activity, but they are relatively more productive in drought conditions.

The average values obtained from the two biological repetitions of the experiment were used to present the results for each of the investigated trophic groups of microorganisms. The primary experimental data and statistical treatment are shown in Appendix A.

### 2.1. Population Levels of Trophic Groups of Microorganisms Performing Common Biogenic Transformations—Micromycetes and Actinomycetes

Micromycetes and actinomycetes represent a relatively small share of the total microbial community in the rhizosphere, but the multifaceted functions they perform in the ecological-trophic niche they occupy make them important indicators of the overall biological state of the rhizosphere soil environment. The representatives of both groups are saprotrophic microorganisms with a powerful enzyme apparatus capable of mineralizing hard-to-degrade forms of organic and mineral compounds. We found a significant decrease in abundance of natural populations at both trophic groups after application of AMF inoculants in this study.

#### 2.1.1. Micromycetes

The influence of both factors on the population density in the group of micromycetes was statistically significant. The impact of the introduction of AMF inoculants was dominant. The statistical assessment of the effect of the “genotype” factor of the total variance was 6.18%, and of the effect of the “AMF” factor, 36.66%. The population density of micromycetes expressed as CFU g^−1^ a. d. soil in rhizospheres at all garden pea cultivars, compared to those not inoculated with AMF, was reduced by about 10^3^–10^4^ CFU g^−1^ a.d.s. (Figure 1a). The significances of the differences were statistically proven according to Student’s test. The confidence level for the “normal” genotype was *p* < 0.01 after inoculations with mAM and pAM. The differences of “afila” genotypes after inoculation with mAM were proven at *p* < 0.05 and with pAM at *p* < 0.10. The differences between noninoculated plants (controls) of the two groups’ pea genotypes were also demonstrated at *p* < 0.10. Between the two versions with AM fungi (mAM and pAM), the differences were not proven. The average population density of micromycetes in the rhizosphere communities formed in the pea varieties assigned to the “normal” genotype was reduced by more than 50% and by more than 40–30% in the pea “afila” type under the influence of both AMF inoculants (Figure 1b). 

#### 2.1.2. Actinomycetes

The inoculation with AM fungi had a proven effect on the population density of the Actinomycetes group. The impact was 23.54% of the total variance. The influence of the “genotype” factor was insignificant. No statistically distinguishable differences in population density between untreated controls from the two different groups of pea genotypes were found. The differences in population density in the plants untreated with AMF were on the order of 10^7^–10^8^ CFU g^−1^ a.d.s. in the direction of reduction (Figure 2a). The differences between AMF-inoculated treatments and untreated controls in both genotype groups of peas at a low level of significance (*p* < 0.10) were proven in the varieties of the “normal” leaf type after treatment with both mycorrhizal products. In the cultivars of the “afila” leaf type, the difference was proven only after treatment with poly-mycorrhizae. In this group of peas, the differences between the two mycorrhizal products tested at a confidence level of *p* < 0.99 were proven. Expressed as a percentage compared to the controls, the recorded reduction in the average population density of the actinomycete group after AM fungi inoculation in the rhizosphere microbiota of the “normal” pea genotype was 65–79%; for the “afila” genotypes, it was 70–79% (Figure 2b).

The established reductions in the population levels of the two trophic groups of microorganisms (micromycetes and actinomycetes) under the influence of the introduction of arbuscular mycorrhizal fungi into the garden pea rhizosphere indicated changes that would affect the general processes of mineralization of organic compounds in the rhizosphere. They also indicated the possibility of competitive or antagonistic relationships between them and the introduced AMF species external to the aboriginal microbial communities.

### 2.2. Population Levels of Trophic Groups of Microorganisms Carryng Out Transformations of Nitrogen Compounds—Ammonifying and Immobilizing Mineral NH_4_-N Bacteria

#### 2.2.1. Ammonifying Bacteria

A proven dominance on the population development of ammonifying microorganisms in the present study was established under the influence of the AMF: the inoculants. The effect was 15.79% of the total variance. The influence of the “genotype” factor was weak, being only 2.76% and not statistically proven. In contrast to the suppressive effect of AMF inoculants in the population levels of the trophic groups of microorganisms considered so far, their influence on the population density of ammonifying microorganisms was stimulating. The increase in numbers was on the order of 10^7^–10^9^ CFU g^−1^ a.d.s. (Figure 3a). Although the influence of the genotype was not proven, the quantitative changes in the “normal” pea genotype were significant. The differences between the treated and untreated treatments in this type of pea were proven at the level of confidence *p* < 0.01 for both AMF products. The differences from the control in the “afila” pea genotype were not proven. There were also no statistically distinguishable differences either between the controls at the two pea genotypes or between the two mycorrhizal products. However, the relative values of the quantitative increases in population density of ammonifying microorganisms in the treated variants, as compared to the untreated ones and expressed as a percentage, increased many times over (Figure 3b).

#### 2.2.2. Immobilizing Mineral Ammonium Nitrogen (NH_4_^+^-N) Bacteria

While the trophic group of ammonifiers mineralizes the nitrogen-containing compounds in the soil and provides plants with nitrogen in forms they can absorb (NH_4_^+^ and NO_3_^−^), the bacteria of the considered trophic group absorb them for their constructive needs, i.e., they are competitors of plants. At the group of mineral NH_4_^+^-N immobilizers, a dominant statistically reliable influence of 17.78% was found to come from the “genotype” factor. The changes in population density under the influence of the introduction of AM-inoculants were insignificant (4.17%). The quantity of immobilizing bacteria increased by about 10^8^–10^9^ CFU g^−1^ a.d.s. in the rhizosphere of the cultivars with the “normal” leaf type (Figure 4a). Differences with untreated controls were demonstrated at a low level of confidence (*p* < 0.10) only in treatments with mono-mycorrhizal product inoculation. Their population density was reduced after treatment with AM-products within 10^8^ CFU g^−1^ a.d.s. in the rhizospheres of the “afila” type varieties. Differences from the control under the influence of both AMF products were proven. The relative values of the differences in the population density of mineral nitrogen-immobilizing bacteria were an increase of between 51.13 and 67.79% in the “normal” genotype and a decrease of 58.76–63.40% in the “afila” type varieties, respectively (Figure 4b).

### 2.3. Population Level of the Trophic Group of Phosphate-Solubilizing Bacteria

Phosphate-solubilizing bacteria (PSB) are a main participant in the transformation of phosphorus compounds in the soil and the conversion of insoluble phosphate into a soluble form that can be absorbed by plants. The importance they have for the phosphorus balance in the soil and the phosphorus nutrition of plants makes them an important type of plant growth-promoting rhizobacteria (PGPRB).

In both tested factors, genotype and AMF had a proven statistical impact on the population density of this trophic group. The effect of applying AMF inoculants was much stronger. The performed variance analysis showed a 51.44% impact of the total variance. The influence of the “genotype” factor was 5.14%. The changes in the population were in the direction of stimulating the numerical development of PSB. The increase was on the order of 10^6^–10^7^ CFU g^−1^ a.d.s. (Figure 5a). Differences in abundance in noninoculated plants at the cultivars with the “normal” leaf type inoculated with a mono-mycorrhizal (mM) product were proven with a high confidence level (*p* < 0.999). The differences between the control variants at the two pea genotypes, “normal” and “afila”, were also statistically proven at a low level of confidence (*p* < 0.10). The reported differences in PSB population density between the other treatments included in the trial were not proven. The relative values of the quantitative increases compared to the control group are represented in Figure 5b.

### 2.4. Population Level of Diazotrophic Bacteria

In our study of the microorganisms carrying out the process of atmospheric nitrogen fixation, we investigated population changes in two groups: free-living nitrogen-fixing bacteria, Azotobacter, and bacteria forming symbiotic relationships with legumes, Rhizobium.

#### 2.4.1. Free-Living Nitrogen-Fixing Bacteria—Azotobacter

The population density of free-living diazotrophic bacteria Azotobacter in all treatments included in the study increased. The impact of both factors was statistically proven, with the dominating effect of the “AMF” factor being 33.66%. The impact of the factor “genotype” was 6.12% of the total variance. The differences between the controls of the two pea genotypes were proven at a low level of confidence (*p* < 0.10). The stimulating effect of the application of AMF-inoculants on the numbers of Azotobacter was in the range of 10^6^–10^7^ CFU g^−1^ a.d.s. (Figure 6a). Quantitative differences compared to noninoculated plants of the “normal” leaf type were demonstrated with a confidence level of *p* < 0.99 with mono-mycorrhizae and with a probability of *p* < 0.95 after application of a poly-mycorrhizae. The percentage increase in the population density of Azotobacter as a relative share compared to the noninoculated treatments was over 60–80% (Figure 6b).

#### 2.4.2. Legume-Symbiotic Nitrogen Fixing Bacteria—Rhizobium

Statistically proven was the influence of the genotype on the group of symbiotic nitrogen-fixing bacteria Rhizobium. The influence of the “AMF” factor was not statistically proven. Differences were observed when comparing average population densities between the two pea types under the influence of AMF inoculation. In cultivars of the “normal” genotype, the number of Rhizobium increased by about 10^5^ CFU g^−1^ a.d.s. The response of Rhizobium populations in the rhizosphere of “afila” cultivars was the opposite; average abundances were reduced (Figure 7a). The differences were statistically reliable according to the genotypes’ respective untreated controls and between the controls themselves and their corresponding genotypes. The difference between the control “normal” group and the control “afila” group was proven. In the varieties of the “normal” leaf type after inoculation with AMF products, the change was evidenced at a level of *p* < 0.95. In the “afila” cultivars, the proven significance levels of the differences were *p* < 0.10 for the treatment with mono-mycorrhiza and *p* < 0.95 for the treatment with poly-mycorrhiza. The relative share of changes in population levels of Rhizobium was an increase of 60–90% in the “normal” genotype and a corresponding decrease in the “afila” genotype by 29–40% (Figure 7b).

### 2.5. Extent of AMF Colonization of Pea Roots

The soil used in our experiment was not sterilized, thus it contained spores from aboriginal AM fungi populations. The established extent of mycorrhization in the roots of the plants grown without the additional introduction of AMF-inoculants was between 10% and 17% in the cultivars with the “normal” leaf type and between 16% and 24% in the varieties with the “afila” leaf type. These percentage extents of mycorrhization were only from the natural mycorrhizal population in the soil. The differences between the two genotypes were statistically proven at *p* < 0.10. The influence of both the “genotype” and “AMF-inoculants” factors were significant at a high level of confidence. The impact of the genotype was 16.29% of the total variance, and the impact of the AMF-inoculants was 46.94%. Additional introduction of AMF into the rhizosphere increased the extent of root colonization. Pea varieties of the “afila” type were distinguished by a higher degree (Figure 8a). Differences between the respective control treatments were proven at a confidence level of *p* < 0.999 after inoculation with a mono-mycorrhizal product and at *p* < 0.99 with a poly-mycorrhizal product in plants of the “normal” leaf type. In the “afila” plants, differences were proven at *p* < 0.99 for the treatment with mAM and at *p* < 0.999 for the treatment with pAM. Between the two mycorrhizal products, the differences were not reliable. The increase in the root mycorrhization, expressed as a relative share compared to that in noninoculated plants, was over 85% (Figure 8b).

### 2.6. General State of the Rhizosphere Microbiota

The impact of the observed changes in the population levels of the considered trophic groups of microorganisms on the structure and functional stability of the communities as a whole was assessed. In the microbial communities, strong-to-moderate correlations were found between the percentage of root colonization with AMF and the population density of the trophic groups of microorganisms except for those of the groups of mineral nitrogen-immobilizing bacteria and Rhizobium bacteria. Dependencies were also registered between individual trophic groups. In Rhizobium bacteria, no relationships were observed with the groups of micromycetes and actinomycetes. In the case of Azotobacter bacteria, no dependence was established only with the group of mineral nitrogen-immobilizing bacteria. In this group (mineral NH_4_^+^-N immobilizing bacteria), a strong positive correlation was registered only with ammonifying bacteria. In the group of ammonifying bacteria, there was no correlation with micromycetes (Table 1).

The ratio between the amounts of mineral nitrogen-immobilizing bacteria and ammonifying bacteria is an indicator of the speed and direction of mineralization processes of nitrogen-containing compounds in the soil. Numerical expressions of the equilibrium states between these two trophic groups in their microbial communities are the mineralization–immobilization index (MII) values. The obtained MII values were greatly underestimated in the AMF-inoculated variants, thus indicating accelerated mineralization. This, in turn, indicates the presence of sufficient absorbable nitrogen in the rhizosphere, which favors the mineral nutrition of the plants (Table 2).

The calculated Shannon diversity index (H) and evenness of distribution (E) in the microbial communities did not show significant deviations from the control variants. Regardless of an outlined tendency to decrease of the values in the variants inoculated with AM fungi, which was more pronounced in the cultivars of the “afila” genotype, they can be classified as moderate. Probably, the decrease in the numerical values of the indices was related to the spatial redistribution in the rhizosphere community because the average population density in them greatly increased by about 10^8^–10^9^ CFU g^−1^ a.d.s. A high degree of correlation was reported between the two indicators characterizing biodiversity at the population level in the rhizosphere microbiota (Table 2).

The hierarchical cluster analysis showed relative similarity among the structures of the microbial communities based on population changes. Most of the trophic groups of microorganisms were united in one group at significantly close distances. Exceptions were the groups of ammonifying and mineral NH_4_^+^-N assimilating bacteria. Subsequent clustering created clusters of the following groups: phosphate-dissolving bacteria, Azotobacter, actinomycetes and mineral NH_4_^+^-N assimilating bacteria. These four trophic groups together with the ammonifying bacteria formed a cluster with the greatest distance. The groups of micromycetes and Rhizobium bacteria remained outside this cluster during the period of this research (Figure 9a). When grouped by pea genotypes, two separate clusters were formed at a relatively short distance corresponding to the “normal” and “afila” types. Subsequently, they formed a common cluster (Figure 9b).

## 3. Discussion

The interactions in the rhizosphere microbiota formed under the influence of the tripartite symbiosis between legumes (*Pisum sativum* L.), Rhizobium and arbuscular mycorrhizal fungi, as mentioned, are multi-layered. Questions regarding the influence of these triple bonds on other members of the rhizosphere microbiota are not sufficiently elucidated [26,27,56]. The used experimental model gave us the opportunity to follow the changes that occur in them after the introduction of AM fungi foreign to the autochthonous communities at the population level under conditions as close as possible to the agroecological habitats of the garden pea. The microbial groups covered in our study include organotrophic microorganisms, which are characteristic inhabitants of natural microcenoses. They are closely related to the general biogenic processes in the soil and the transformations of carbon, nitrogen and phosphorus compounds. The assignment of microorganisms to a given trophic group is based on the functions they perform in a rhizosphere community as determined by their energy and constructive metabolism [43]. The specificity, dynamics in the structure and ecological-trophic relationships both between individual groups of microorganisms and between them and plants are determined by the course of metabolic processes.

Here, we established a diverse population response specific to each of the considered trophic groups of microorganisms to the introduction of AM fungi into the pea rhizosphere. We registered a strong decrease in the population levels of micromycetes and actinomycetes, which was related to the mineralization activity of hard-to-degrade organic compounds in the soil. The reason may be the occurrence of competitive or antagonistic relationships with AM fungi. Although the proliferation of AM fungal hyphae in organic materials has been reported [58], their direct involvement in the mineralization of organic nutrients is questionable. For their metabolism, AM fungi obtain carbon from the host plant [12]. In our opinion, the more likely reason for the reduced population density in the micromycete and actinomycete populations was an antagonistic effect by the inoculated AM fungi. Our results were in line with the established suppressive effect of AM fungi on the development of some fungal phytopathogens [11,14]. They are actually part of the natural population of micromycetes in the rhizosphere. The observed impact may be an effect of the improved health and immune status of the mycorrhized plants, but data were accumulating about the presence of compounds in the chemical composition of the hyphal exudates released by AM fungi in the rhizosphere that can explain the presence of antifungal and, in some cases, antibacterial effects [13,18,59]. According to the data obtained by us on the population changes in the two trophic groups involved in the mineralization and immobilization of nitrogen-containing compounds in the soil, the participation of AM fungi in the mineralization processes was rather indirect. The basis for this opinion was the stimulating effect of AMF exudates on bacterial growth established by other authors, as the mechanisms for this effect are not yet fully understood [60]. It has been suggested that the effect may be due to increased carbon supply, chemotaxis and/or signaling molecules stimulating bacterial gene expression that controls the processes of the “quorum sensing” phenomenon in these bacteria [50]. The increased population levels of ammonifying bacteria we observed may also be due to a synergistic effect on the part of AM fungi as they absorb NH_4_^+^ ions released by the ammonifying activity of these bacteria. The lack of a trophic relationship with the group of mineral nitrogen-immobilizing bacteria can explain the indifferent attitude of AM fungi on the population density of this group of microorganisms.

Extraradical hyphae of AM fungi have been shown to retain and transport bacteria across considerable distances to food stains [27]. This function of theirs is mainly accomplished by forming a carbohydrate-rich aqueous biofilm around the hyphae which, in addition to transport, provides the bacteria with the necessary metabolic energy [14]. This transport mechanism has been demonstrated in phosphate-solubilizing bacteria [61]. Between AM fungi and alkaline phosphatase-producing bacteria, mutual relationships were established, which was one of the reasons for increasing the mineralization of organic phosphorus compounds. The positive impact of AM-mycorrhiza on the phosphorus nutrition of plants was observed [13,24]. The same authors hypothesized that AM fungi control the interaction with phosphorous bacteria by regulating phosphate homeostasis [61]. It was possible with this type of transport to use the ammonifying bacteria, especially because the functions performed by the two trophic groups in the mineralization process of organic substances overlap to a certain extent. Members of the ammonifying group may also be members of the phosphate-solubilizing bacteria group. Our results were in agreement with the established stimulating role of AM fungi on the abundance of phosphate-solubilizing bacteria. A very strong increase in the number of phosphate-dissolving bacteria was registered under the influence of the applied mycorrhizal inoculants.

The other trophic group of microorganisms strongly affected by the inoculation of AM fungi in our study was the group of free-living diazotrophic bacteria Azotobacter. A synergistic effect of Azotobacter and AM fungi on plant development has been reported [24,25]. In our opinion, an explanation for the high population density found in this group was the background enriched in absorbable nitrogen NH_4_^+^-N and absorbable phosphorus as a result of increased ammonification and phosphate mineralization. This was also supported by the registered positive correlations between the population of Azotobacter, in addition to the degree of mycorrhization, and the densities in both trophic groups.

Interestingly, no effect of AMF inoculants was found on the numerical development of the group of symbiotic diazotrophic bacteria Rhizobium in the present study. Some authors reported positive synergistic or mutualistic relationships [24,26], but other authors found none [25]. Our results obtained in the specific conditions of the experiment support the claim of no positive synergy between Rhizobium bacterial population density in the rhizosphere and AMF root colonization in the garden pea. In fact, the results obtained from this study showed that the leading factor for the formation of natural rhizobial populations as well as populations from the trophic group of NH_4_^+^-N immobilizing bacteria were the pea genotypes. The distinctive specificity of legume–rhizobia symbiosis was known for the formation of symbiotic relationships only between specific types of legume and bacterial species, including specificity between cultivars and Rhizobium strains [22]. We also found differences in the rhizosphere microbiota between individual pea genotypes in both the Rhizobium group and the mineral nitrogen-immobilizing bacteria group.

Root exudates, along with nutrient flow and soil composition, are known to be the determinants of microbiota formation [62]. Hormonal effects on the part of plants are also involved in the rhizosphere relations. Plants have been shown to regulate the balance of establishing symbiotic relationships by autoregulating their hormone levels. Specific changes in the hormonal profile under the influence of mycorrhization in other cultures have been recorded.

Already mycorrhized plants have been found to secrete fewer signaling molecules of the strigolactone type required for AMF spore germination [30,63,64]. Our results showed a higher degree of AMF colonization of the roots of plants of the “afila” genotype and a decrease in the population density of Rhizobium bacteria. This fact correlates with reports of a decrease in the number of rhizobial tubers with increased mycorrhization [65]. Hormonal autoregulatory mechanisms likely account for the different responses we found between individual pea genotypes “normal” and “afila” to population formation by symbiotic Rhizobium bacteria and the degree of colonization by AM fungi.

Most of the studies on AM symbiosis in the pea crop have been carried out with the species *Rhizophagus irregularis* (*Glomus intraradices*) [5,6,26]. This species was present in both bioproducts we used. We are not aware of studies with other species of AM fungi mixed in the poly-mycorrhizal product. The reasons for the weak differences and not-always distinct trends in the population response of the microbial communities to the two mycorrhizal products registered in the present study may be of a different nature: weaker adaptation to the microbial community of the AMF species included in the poly-mycorrhizal bioproduct, relationships between the AMF species themselves inside the AM mix, silencing or suppressing the impact of *R. irregularis*, etc. Future research with individual species of AM fungi is needed to clarify and find answers.

In conclusion, the experimental model we used with different garden pea genotypes inoculated with AM fungi gave us the opportunity to establish their influence on the population levels of major trophic groups of soil microorganisms and the complex population response of the rhizosphere microbiota. We found a dominant effect of AM fungi on micromycete and actinomycete densities in the direction of reduction, which suggests antagonistic relationships. We determined the formation of mutualistic relationships between AMF and ammonifying, phosphate-solubilizing and free-living diazotrophic bacteria Azotobacter. We found that the type of genotype was decisive for the formation of the populations of mineral ammonium nitrogen-immobilizing bacteria and Rhizobium bacteria. We reported significant two-way dependencies between trophic groups of microorganisms whose vital activities are related to the transformations of nitrogen and phosphorus compounds in the soil. The registered changes in overall abundance in the microbial communities, the distribution between individual trophic groups and the general population response of the rhizosphere pea microbiota do not indicate changes in the structural and functional stability of the communities. Our findings add to the knowledge on the complex system of microbial relationships in the rhizosphere microbiota in the pea tripartite symbiosis and may be useful in future studies in combination with metagenomic and metabolomic studies.

## 4. Materials and Methods

Plant materials: Six garden pea (*Pisum sativum* L.) cultivars from the working collection of the Maritsa Vegetable Crop Research Institute were selected for objects. Three of them (Marsi-n., Plovdiv-n., 22/16-n.) were of the “normal” leaf type, and three (Echo-af, Kasino-af, 22/16-af) were of the “afila” leaf type. Photos of garden pea genotypes with “normal” and “afila” leaf types are shown in Appendix A.

Arbuscular mycorrhizal fungi: AM fungi were purchased as ready-to-use inoculants:MycoPlant^®^ from Tratamientos Bio-Ecológicos, San Javier, Spain, (PE-3536). The product contains spores (≥400.g^−1^) of one species of AM fungus *Rhizophagus irregularis* (*Glomus intraradices*).FunkyFungi® B.A.C. from Gouda, the Netherlands, (PE-844566). The product contains spores (≥220∙g^−1^) of four species of AM fungi: *Acaulospora colombiana* (*Entrophosphora colombiana), Claroideoglomus etunicatum (Glomus etunicatum), Rhizophagus clarum (Glomus clarum)* and *R. intraradices* (*G. intraradices)*.

Experimental design and growth conditions: The study was conducted during two phenological periods of garden pea, in pot experiments, under greenhouse conditions at the Maritsa Vegetable Crop Research Institute, Plovdiv, Bulgaria (42°10’35.3” N 24°45’50.5” E). Pots were arranged in a randomized complete block design with three replications of treatments. The experimental pots had a capacity of 12 L, filled with alluvial-meadow type soil, Fluvisols (FAO), and with the following agrochemical characteristics: organic matter content (by Tyurin): 1.9%; total nitrogen content (by Kjeldahl): 1.7 mg/kg^−1^; available P_2_O_5_: 21.4 mg/kg^−1^ (by Egner–Reem); available K_2_O: 22.4 mg/kg^−1^ (in 2n HCl); soil pH (H_2_O): 6.44; total soluble salts: 0.13 mg/cm^−1^ by electrical conductivity (EC mS/cm).

Four plants of each variety were grown in a pot. AMF-inoculants were introduced into the soil substrate during the sowing of peas (March 15th) at a depth of 5 cm in doses of 2 g per plant. AM-inoculants were applied as solid granules. The pots were watered regularly every 3–4 days with tap water. Greenhouse growing conditions were at temperature 22/16 °C and 65–70% humidity. In the period of the experiments, the durations of day and night were 12–14h per day and 12–10 h per night.

The experiment included 18 treatments. For each of the six garden pea genotypes, the following variants were formed: (1) Control (untreated); (2) Mono-mycorrhizal (mAM) treatment; (3) Poly-mycorrhizal (pAM) treatment.

### Microbiological Analyses

The microbiological analyses were carried out on soil samples taken from the rhizosphere zone during the flowering period of the plants (about 35–40 days after sowing). The plants were removed from the soil. Average subsamples of the soil adhering to the plant roots from all replicates of each treatment were prepared. Subsamples included all soil (about 5–8 mm) from the rhizosphere continuum, including the ectorrhizosphere, rhizoplane and root endorhizosphere. Soil samples (5 g) were dried to a constant weight at 105 °C, the amount of evaporated water was determined, and the moisture content (%) and the amount of dry matter per g of soil were calculated, which were used in the formula for calculating the amount of microorganisms per units forming colonies per g of absolutely dry soil (CFU/g a.d.s.).

The quantitative microbiological analyses were performed according to Koch’s method by cultivating 10-fold diluted soil suspensions on specific nutrient media for each trophic group of microorganisms [66]. Analyses were carried out in three replications. The quantities were defined after an incubation period at 28 °C. The population sizes were estimated as colony-forming units per g absolutely dry soil (CFU/g a.d.s.), with confidence level 0.05 the following formula: $\mathrm{CFU}=\frac{\left(\;\bar{\chi }\pm {t\sigma }_{\chi }\right).\frac{K}{V}\;}{D_{m}}$ where $\;\bar{\chi }$ is the average number of colonies of all reps; t = 2 at P_0.95_; σ_χ_ is the mean square deviation; K is the dilution from which has been made; V is the the volume of the inoculum in ml; D_m_ is the amount of dry matter in g soil. The population densities at the following trophic groups of pea soil-born rhizosphere microorganisms were determined:

Micromycetes were incubated on Czapek’s agar (CZA) medium for 7 days; actinomycetes were incubated on Gause’s synthetic agar medium (GSA) for 7 days; ammonifying bacteria were incubated on Meat—Peptone stock agar (MPA) medium for 5 days; mineral NH_4_^+^-N immobilizing bacteria were incubated on starch–ammonium mineral salts (SAMA) agar medium for 7 days; Phosphate-solubilizing bacteria were incubated on Pikovskaya’s agar medium for 10 days (only colonies that formed halo zones were counted); Azotobacter bacteria were incubated on Ashby’s Mannitol agar N_2_-free medium for 7 days; Rhizobia bacteria were incubated on Yeast Mannitol Agar w/Congo Red for 48 h (only colonies typical of Rhizobia, which do not absorb the red color on the YEMA medium, were counted).

Isolation nutrient media compositions are shown in Appendix A.

The ratio between the quantities of NH_4_^+^-N mineral-immobilizing bacteria and the quantities of ammonifying bacteria was calculated, and it determined the values of the mineralization–immobilization index (MII). The diversity indexes of Shannon (H) and Simpson (D) and the distribution evenness (E_H_) in the microbial communities were determined. Calculations were made using the following equations:$\;H=\overset{s}{\underset{i=1}{\sum }}p_{i}{\mathrm{lnp}}_{i}$ and E_H_ =$\;\frac{H}{\mathrm{lnS}}$ [67]. We conditionally accepted the abundance of the individual trophic groups of microorganisms as functional units in the microbial communities formed in their entirety at the individual treatments.

AMF detection: AMF colonization (%) was evaluated from random subsamples of root segments of plants from each variant. The roots, after being washed, were cleared in 2% KOH (90 °C) for 45 min, acidified in 1% HCl for 15 min and stained with 0.05% Trypan Blue in acid glycerol (90 °C) for 45 min, according to the procedure of Koske and Gemma [68]. The percentage of total root colonization (arbuscules, vesicles and hyphae) was determined with the gridline intersection method [69]. The readings were carried out with a binocular microscope (magnification 4×) with 100 root pieces.

Statistical analysis: The data were subjected to a two-factor analysis of variance. The effects (ηx^2^) of the factors of AM fungi and pea genotype as well as the levels of their statistical significance (*p*) were determined according to Fisher’s test (F). The significance of differences between means of controls and treatments was determined with the two-tailed Student’s *t*-test. Correlation between amounts of the trophic groups’ microorganisms with root colonization were determined with AMF (where r is Pearson’s coefficient). Hierarchical cluster analysis of the structure of microbial communities was conducted using Ward’s method. Statistical analyses were performed using the program package SPSS 13.

## Figures and Tables

**Figure 1 ijms-24-01119-f001:**
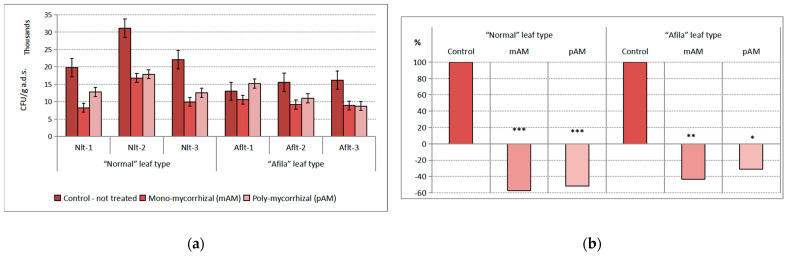
Average population density of the trophic group Micromycetes (CFU g^−1^ absolutely dry soil) in the rhizosphere communities formed after introduction of AM fungi: (mAM) mono-mycorrhizal and (pAM) poly-mycorrhizal inoculants. (**a**) Average values for the particular varieties of garden pea with “normal” leaf type (Nlt-1—cvr. Marsi, Nlt-2—cvr. Plovdiv and Nlt-3—cvr. 22/16-n) and “afila” leaf type (Aflt-1—cvr. 22/16-af, Aflt-2—cvr. Kazino and Aflt-3—cvr. Echo). Error bars indicate standard error of the mean (SE). (**b**) Relative values of the differences in the population levels of the trophic group Micromysetes (%) compared to the untreated controls, including means for both “normal” leaf type and “afila” leaf type genotypes. The significance of the differences between means of controls and treatments were calculated with the two-tailed Student’s *t*-test (at df—10; *** *p* < 0.01, ** *p* < 0.05 and * *p* < 0.10).

**Figure 2 ijms-24-01119-f002:**
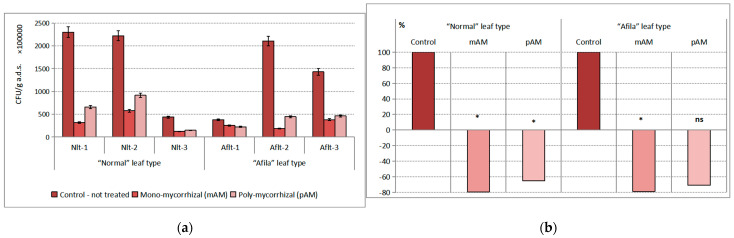
Average population density of the trophic group Actinomycetes (CFU g^−1^ absolutely dry soil) in the rhizosphere communities formed after introduction of AM fungi: (mAM) mono-mycorrhizal and (pAM) poly-mycorrhizal inoculants. (**a**) Average values for the particular varieties of garden pea with “normal” leaf type (Nlt-1—cvr. Marsi, Nlt-2—cvr. Plovdiv and Nlt-3—cvr. 22/16-n) and “afila” leaf type (Aflt-1—cvr. 22/16-af, Aflt-2—cvr. Kazino and Aflt-3—cvr. Echo). Error bars indicate standard error of the mean (SE). (**b**) Relative values of the differences in the population levels of the trophic group Actinomycetes (%) compared to the untreated controls, including means for both “normal” leaf type and “afila” leaf type genotypes. The significance of the differences between means of controls and treatments were calculated with the two-tailed Student’s *t*-test (at df—10; * *p* < 0.10), ns—nonsignificant).

**Figure 3 ijms-24-01119-f003:**
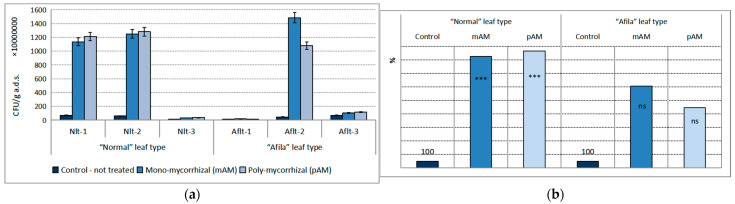
Average population density of the trophic group Ammonifying bacteria (CFU g^−1^ absolutely dry soil) in the rhizosphere communities formed after introduction of AM fungi: (mAM) mono-mycorrhizal and (pAM) poly-mycorrhizal inoculants. (**a**) Average values for the particular varieties of garden pea with “normal” leaf type (Nlt-1—cvr. Marsi, Nlt-2—cvr. Plovdiv and Nlt-3—cvr. 22/16-n) and “afila” leaf type (Aflt-1—cvr. 22/16-af, Aflt-2—cvr. Kazino and Aflt-3—cvr. Echo). Error bars indicate standard error of the mean (SE). (**b**) Relative values of the differences in the population levels of the trophic group Ammonifying bacteria (%) compared to the untreated controls, including means for both “normal” leaf type and “afila” leaf type genotypes. The significance of the differences between means of controls and treatments were calculated with the two-tailed Student’s *t*-test (at df—10; *** *p* < 0.01, ns—nonsignificant).

**Figure 4 ijms-24-01119-f004:**
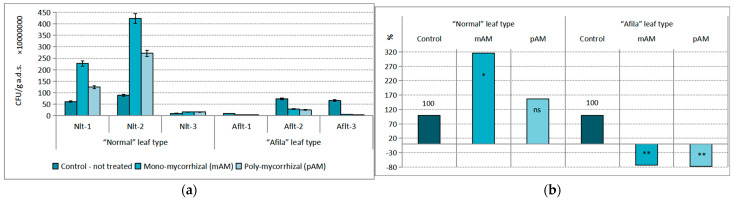
Average population density of the trophic group Immobilizing mineral ammonium nitrogen (NH_4_^+^-N) bacteria (CFU g^−1^ absolutely dry soil) in the rhizosphere communities formed after introduction of AM fungi: (mAM) mono-mycorrhizal and (pAM) poly-mycorrhizal inoculants. (**a**) Average values for the particular varieties of garden pea with “normal” leaf type (Nlt-1—cvr. Marsi, Nlt-2—cvr. Plovdiv and Nlt-3—cvr. 22/16-n) and “afila” leaf type (Aflt-1—cvr. 22/16-af, Aflt-2—cvr. Kazino and Aflt-3—cvr. Echo). Error bars indicate standard error of the mean (SE). (**b**) Relative values of the differences in the population levels of the trophic group Immobilizing mineral ammonium nitrogen (NH_4_^+^-N) bacteria (%) compared to the untreated controls, including means for both “normal” leaf type and “afila” leaf type genotypes. The significance of the differences between means of controls and treatments were calculated with the two-tailed Student’s *t*-test (at df—10; ** *p* < 0.05 and * *p* < 0.10, ns—nonsignificant).

**Figure 5 ijms-24-01119-f005:**
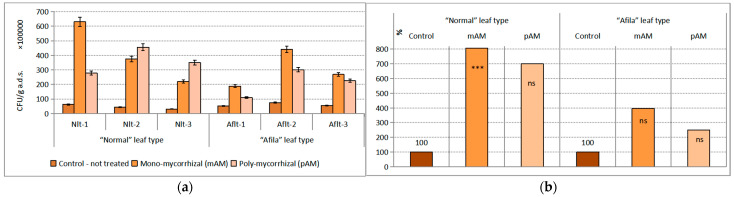
Average population density of the trophic group Phosphate-solubilizing bacteria (PSB) (CFU g^−1^ absolutely dry soil) in the rhizosphere communities formed after introduction of AM fungi: (mAM) mono-mycorrhizal and (pAM) poly-mycorrhizal inoculants. (**a**) Average values for the particular varieties of garden pea with “normal” leaf type (Nlt-1—cvr. Marsi, Nlt-2—cvr. Plovdiv and Nlt-3—cvr. 22/16-n) and “afila” leaf type (Aflt-1—cvr. Marsi, Aflt-2—cvr. Plovdiv and Aflt-3—cvr. 22/16-af). Error bars indicate standard error of the mean (SE). (**b**) Relative values of the differences in the population levels of the trophic group Phosphate-solubilizing bacteria (PSB) (%) compared to the untreated controls, including means for both “normal” leaf type and “afila” leaf type genotypes. The significance of the differences between means of controls and treatments were calculated with the two-tailed Student’s *t*-test (at df—10; *** *p* < 0.01, ns—nonsignificant).

**Figure 6 ijms-24-01119-f006:**
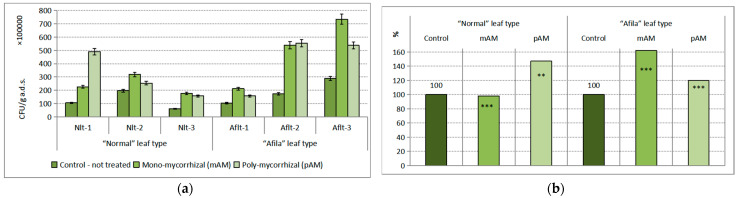
Average population density of the free-living diazotrophic bacteria Azotobacter (CFU g^−1^ absolutely dry soil) in the rhizosphere communities formed after introduction of AM fungi: (mAM) mono-mycorrhizal and (pAM) poly-mycorrhizal inoculants. (**a**) Average values for the particular varieties of garden pea with “normal” leaf type (Nlt-1—cvr. Marsi, Nlt-2—cvr. Plovdiv and Nlt-3—22/16-n) and “afila” leaf type (Aflt-1—cvr. 22/16-af, Aflt-2—cvr. Kazino and Aflt-3—cvr. Echo). Error bars indicate standard error of the mean (SE). (**b**) Relative values of the differences in the population levels of the free-living diazotrophic bacteria Azotobacter (%) compared to the untreated controls, including means for both “normal” leaf type and “afila” leaf type genotypes. The significance of the differences between means of controls and treatments were calculated with the two-tailed Student’s *t*-test (at df—10; *** *p* < 0.01 and ** *p* < 0.05).

**Figure 7 ijms-24-01119-f007:**
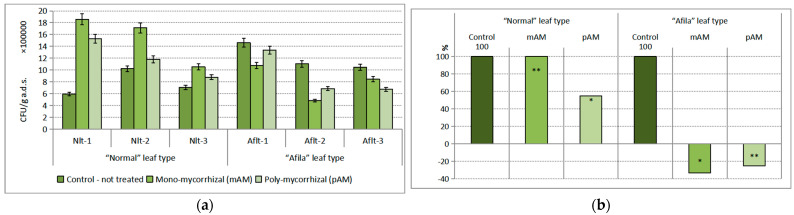
Average population density of the symbiotic nitrogen-fixing bacteria Rhizobium (CFU g^−1^ absolutely dry soil) in the rhizosphere communities formed after introduction of AM fungi: (mAM) mono-mycorrhizal and (pAM) poly-mycorrhizal inoculants. (**a**) Average values for the particular varieties of garden pea with “normal” leaf type (Nlt-1—cvr. Marsi, Nlt-2—cvr. Plovdiv and Nlt-3—cvr. 22/16-n) and “afila” leaf type (Aflt-1—cvr. 22/16-af, Aflt-2—cvr. Kazino and Aflt-3—cvr. Echo). Error bars indicate standard error of the mean (SE). (**b**) Relative values of the differences in the population levels of the symbiotic nitrogen-fixing bacteria Rhizobium (%) compared to the untreated controls, including means for both “normal” leaf type and “afila” leaf type genotypes. The significance of the differences between means of controls and treatments were calculated with the two-tailed Student’s *t*-test (at df—10; ** *p* < 0.05 and * *p* < 0.10).

**Figure 8 ijms-24-01119-f008:**
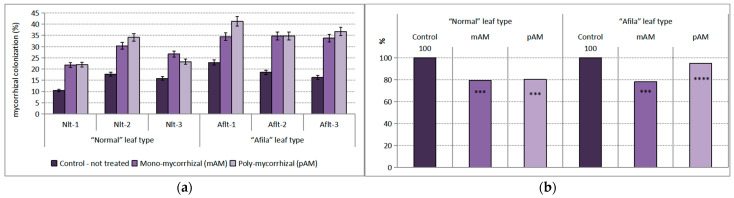
Degree of mycorrhizal colonization (%) of garden pea genotypes roots. (**a**) Average values for the particular varieties of garden pea with “normal” leaf type (Nlt-1—cvr. Marsi, Nlt-2—cvr. Plovdiv and Nlt-3—cvr. 22/16-n) and “afila” leaf type (Aflt-1—cvr. 22/16-af, Aflt-2—cvr. Kazino and Aflt-3—cvr. Echo). Error bars indicate standard error of the mean (SE). (**b**) Relative values of the differences in the percent of colonization (%) compared to the untreated controls, including means for both “normal” leaf type and “afila” leaf type genotypes. The significance of the differences between means of controls and treatments were calculated with the two-tailed Student’s *t*-test (at df—10; **** *p* < 0.001 and *** *p* < 0.01).

**Figure 9 ijms-24-01119-f009:**
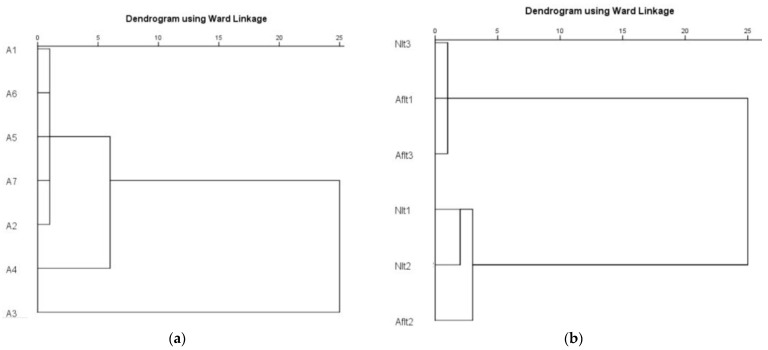
Hierarchical cluster analyses of rhizospherae microbial communities in garden peas. (**a**) Analysis by trophic groups of microorganisms. A1: Micromycetes; A2: Actinomycetes; A3: Ammonifying bacteria; A4: Immobilizing mineral NH_4_^+^-N bacteria; A5: Phosphate-solubilizing bacteria; A6: Azotobacter bacteria; A7: Rhizobium bacteria. (**b**) Analysis by pea genotypes. Cultivars of garden pea with the “normal” leaf type (Nlt-1—cvr. Marsi, Nlt-2—cvr. Plovdiv and Nlt-3—cvr. 22/16-n) and “afila” leaf type (Aflt-1—cvr. 22/16-af, Aflt-2—cvr. Kazino and Aflt-3—cvr. Echo) (Ward’s method was used).

**Table 1 ijms-24-01119-t001:** Correlation (where r is Pearson’s coefficient) between the quantities of the trophic groups of microorganisms with the percentage of AM-colonization of garden pea roots in the microbial communities.

AMF	AMF	MmS	AcS	AmB	ImNB	PSB	AzB	RhB
	1							
MmS	−0.585 **	1						
AcS	−0.635 ***	0.768 ***	1					
AmB	0.282 *	−0.161 ns	−0.222 *	1				
ImNB	−0.024 ns	0.165 ns	0.117 ns	0.634 ***	1			
PSB	0.398 **	−0.513 **	−0.479 **	0.774 ***	0.516 **	1		
AzB	0.513 **	−0.452 **	−0.264 *	0.428 **	−0.003 ns	0.383 **	1	
RhB	0.016 ns	−0.103 ns	−0.136 ns	0.260 *	0.634 ***	0.293 *	−0.233 *	1

Abbreviations: AMF: Arbuscular mycorrhizal fungi; MmS: Micromycetes; AcS: Actinomycetes; AmB: Ammonifying bacteria; ImNB: Immobilizing mineral NH4+-N bacteria; PSB: Phosphate-solubilizing bacteria; AzB: Azotobacter bacteria; RhB: Rhizobium bacteria. Significance of r, Pearson’s coefficient: strong *** r ≥ 0.660; medium ** r ≥ 0.330–0.660; low * r ≥ 0.220–0.330; nonsignificant ns *p* < 0.220.

**Table 2 ijms-24-01119-t002:** Average population density in the microbial communities (CFU g^−1^ a.d.s./log) and values of the calculated indices to assess the state of homeostasis, distribution evenness and richness in the rhizosphere microbiota of garden peas after inoculation with AM fungi.

Indices	“Normal” Leaf Type			“Afila” Leaf Type		
	Control	mM	pM	Control	mM	pM
MPD	8.225	9.171	9.152	8.179	8.902	8.783
MII	1.364	0.105	0.099	1.339	0.337	0.248
Shannon (H)	1.939	1.338	1.244	1.961	0.552	0.669
E	0.996	0.688	0.639	1.008	0.284	0.344
r	0.985			0.976		

Abbreviations: mM: mono-mycorrhizal; pM: poly-mycorrhizal; MPD: mean population density/log; MII: mineralization–immobilization index; H: Shannon’s diversity index; E: distribution evenness; r: Pearson’s coefficient.

## Data Availability

The data described in this study can be found in the article and the Appendix A. The seed materials are available upon request from the author Tsveta Hristeva.

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
