# Peer review of "Population Response of Rhizosphere Microbiota of Garden Pea Genotypes to Inoculation with Arbuscular Mycorrhizal Fungi"

_ijms, 2023, doi:10.3390/ijms24021119_

Round 1

Reviewer 1 Report

More information on the interaction of beneficial microbes (such as AM fungi) and the soil microflora is needed to increase our knowledge on possible positive effects for plant health and growth. Therefore, experiments such as the presented ones should be encouraged. The authors present a rather limited set of experiments with some interesting results. However, the manuscript could be improved in several regards, and the results are not always very clear.

I had major problems to understand some of the results presented in Figures 1 to 8 and found them misleading, but maybe that’s because I didn’t look at them in the right way??

Figures 1-8 present data on the average population density of the different trophic groups (a), and the relative values of the differences after adding of mM and pM inoculants. In general, I like this kind of graphic presentation, it is very clear and easy to follow. However, it seems in several cases that the graphs don’t lign up. E.g. Figure 6. The CFU of the non-treated control “normal” range from approx. 0.5x10E7 to 2x10E7. The respective values for mM are approx. 2-3x10E7. But on Graph b a relative value for mM of less than 75% is shown, indicating a decrease of bacterial CFUs after inoculation. Similar discrepancies can be found in other graphs.

In addition, several other points should be addressed:

Line 624-625: More information on the inoculation should be given. Was the AMF-inoculant used as a solid granule or as a solution? Was it added directly to the root system or to a general ‘soil layer’?

Line 631-632:  the soil was analyzed just once (during the flowering period). It is not clear, why this moment was chosen. In general, several analyses would be preferable to get a sense of the dynamics of changes of soil microorganisms over time. If only one analyses is done, I would do it at harvest time, to see if a correlation between microbial numbers and crop weight can be found. It is not clear, how much soil is used for the analysis, and the area ‘adhering to the plant roots’ is not well defined. In addition, it would have been interesting to test also soil from in-between the plants, to see if the inoculation also has an effect on microbial composition of soil farther away from the rhizosphere.

Line 638: I don’t know why the  incubation was done at 28C. This seems to be rather warm for soil analysis, and could favor the growth of some organisms over others and therefore change the results. Maybe the soils which are used in the author’s country for growing peas are rather warm, but it should be explained.

Line 639: it should be explained how the soil was dried.

Line 634: The hierarchical cluster analyses using Ward’s method has to be explained in more details. Which raw data were used for the analysis?

Line 658 and Table 2: The Shannon Index normally is a used to compare diversity of species. In this analysis, species were not identified, only trophic groups. Please explain which criteria you used to calculate the Shannon Index.

Line 585: I think you confused ‘macromycetes’ with ‘micromycetes’

Line 590-591. The authors write: ‘We reported…related to the availability of nitrogen and phosphorus ions in the soil.’ However, the authors did not measure nitrogen and phosphorus levels, therefore this is a hypothesis and not a fact and should be explained as such.

A chemical analysis of the soil to test the effects of AMs on nutrient levels would be interesting for a better understanding of changes of the population levels of the different trophic groups.

Possible effects on non-culturable microorganisms should at least be discussed briefly.

Reviewer 2 Report

The theme of the article is interesting however the quality of work and presentation is not satisfactory. Author had performed very traditional method CUF/g a.d.s. to analyze the microbial density  after so much advancement in technology.  The results output  of plate counting is very basic  and not recommendable to published in such a reputed journal.  

Reviewer 3 Report

The language and grammar of the manuscript may be improved by careful reading. Typographic errors must be taken care of.

Reviewer 4 Report

The research about population response of rhizosphere microorganisms of garden pea genotypes to inoculation with arbuscular mycorrhizal fungi is well done and well performed. Results are interesting and somehow are expected.

However I have some comments:

Line 5 – check the style

In my own opinion, Introduction is too large. Some parts might be shorter.

Line 46 – the talk was about Rhizobia and then suddenly about AM. This looks weird. Also there is too much information about physiological processes of symbiosis in introduction, because the research is about relationship between rhizosphere microorganisms and AMF.

mM and pM – I think it is better to change it with mAM (because it is arbuscular mycorrhiza) and pAM. Seeing mM readers will think of concentration.

Lines 166- 172 (and the same for all parts of results) – the information about statistical tests, that were used, is better to give in Materials and methods. Moreover all this information about p value is unnecessary here. It is given in a figure. It is not good to give a lot of numbers in a text. It means that you cannot perform your results with means of tables or figures and can’t explain your results with words.

What is Nlt-1, Aflt-1? I did not find the information. How can I understand what the cultivar Nlt-1 is?

Line 441- it should be Table 2

Discussion is good and comprehensive. However there are very few words about influence of AMF on different genotypes of pea (lines 564 -566). This is too little. It would be good to add some details. The use of dividing these genotypes in two groups is unclear. Also there are no thoughts and suggestions about influence of different type of mycorrhiza inoculation on rhizosphere. Was it so necessary to perform experiments with mono- and poly- inoculation? According to your results, there is no big difference. However in some cases there is difference. But this is hidden and unclear. Also it should be discussed.

The discussion is rather big now. In my opinion it is possible to shorten the beginning of it.

Line 625 – what is 15.03?

Line 626 – add hours of day/night

Round 2

Reviewer 2 Report

Author have made significant changes in the article. Although the article deals a very basic study but can be accepted in the present form